# Recruiting hard to reach populations to studies: breaking the silence: an example from a study that recruited people with dementia

Becky Field [1], Gail Mountain,[1,2] Jane Burgess,[3] Laura Di Bona,[1,4] Daniel Kelleher,[5] Jacqueline Mundy,[4] Jennifer Wenborn[3,6]

For numbered affiliations see end of article.

**Correspondence to**
Becky Field;
b.field@sheffield.ac.uk

## ABSTRACT

**Objective** To share the challenges of recruiting people with dementia to studies, using experiences from one recently completed trial as an exemplar.

**Background** Research publications always cite participant numbers but the effort expended to achieve the sample size is rarely reported, even when the study involved recruiting a hard to reach population. A multisite study of a psychosocial intervention for people with dementia illustrates the challenges. This study recruited 468 'dyads' (a person with dementia and a family carer together) from 15 sites but the time taken to achieve this was longer than originally estimated. This led to a study extension and the need for additional sites. Recruitment data revealed that certain sites were more successful than others, but why? Can the knowledge gained be used to inform other studies?

**Methods** Secondary analysis of routinely collected recruitment data from three purposefully selected sites was examined to understand the strategies used and identify successful approaches.

**Findings** At all three sites, the pool of potential recruits funnelled to a few participants. It took two sites 18 months longer than the third to achieve recruitment numbers despite additional efforts. Explanations given by potential participants for declining to take part included ill health, reporting they were 'managing', time constraints, adjusting to a diagnosis of dementia and burden of study procedures.

**Conclusions** Successful recruitment of people with dementia to studies, as one example of a hard to reach group, requires multiple strategies and close working between researchers and clinical services. It requires a detailed understanding of the needs and perspectives of the specific population and knowledge about how individuals can be supported to participate in research. Experiences of recruitment should be disseminated so that knowledge generated can be used to inform the planning and implementation of future research studies.

## INTRODUCTION

Research publications report numbers of participants and usually numbers screened and excluded. Yet the effort expended to achieve required sample sizes is rarely reported even when studies recruited hard to reach populations. We contend that transparency about the challenges involved in recruiting hard-to-access populations and potential solutions to the challenges is required to enable future clinical studies to plan and recruit in a time-efficient and cost-effective manner.

Dementia research is a global clinical and research priority.[1 2] In England, it has been proposed that to meet future study requirements, the number of people with dementia participating in dementia research should increase from 4.5% of those diagnosed with dementia to 10%.[3] Yet, it is well documented that people with dementia are a hard to reach population and recruiting the numbers needed for research is challenging.[4–7] This is particularly so for psychosocial research which requires the participant with dementia and often a family carer to consent to possible involvement in an intervention aimed at both people. These studies, like the example used in this article, can be perceived as being particularly demanding for potential recruits.

The example we use here is based on recruitment to one study ('Valuing Active Life in Dementia' (VALID)). VALID first adapted and developed an occupational therapy intervention for community-dwelling people with dementia and their family carers (dyads). The intervention aimed to facilitate independence, meaningful activity, quality of life for the person and carer competence. VALID then evaluated the clinical and cost-effectiveness of the intervention compared with usual care. All participants were asked to complete validated instruments at baseline, 3-month and 6-month follow-up. This involved each person completing questionnaires at home, with a researcher. The intervention involved up to 10 home or

community-based sessions with dyads. These involved the dyad working together with an occupational therapist to identify personal goals and practising suggested strategies to achieve them. Further details of the VALID study are described elsewhere.[8]

Study inclusion criteria involved recruiting the dyad. The time taken to achieve the target sample (n=480) was longer than originally estimated and contributed towards a study extension and the resource-intensive requirement to recruit further sites. Over the course of the study, it became evident that certain of the 15 sites were more successful at achieving recruitment targets than others. As researchers involved in the management and delivery of this study, we wanted to identify the reasons for this.

The objective of this paper is to share the challenges of recruiting people with dementia to studies, using our experiences from the VALID study[8] as an exemplar.

## METHODS

A secondary analysis of recruitment data routinely collected by 3 of 15 participating sites was conducted to compare differences and similarities between recruitment at these sites, the strategies used to promote recruitment and the outcomes of such strategies. The three sites (A, B and C) were selected because they had participated in the VALID pilot study which indicated a substantial pool of potentially eligible participants reportedly available at each site and they had the resources available to support this secondary analysis. Anonymised information was extracted from data routinely collected at each of these sites via 'trial management logs'. Sites A and B used EXCEL for this purpose, site C site used their own, existing recruitment database. Each site collected core information to enable screening for study inclusion criteria. This included records of all contacts with potential recruits during screening and recruitment and those made following recruitment for the duration of the participants' involvement in the study. Researcher notes, which recorded the reasons provided by potential participants for accepting, declining or ineligibility were also analysed and categorised. The data were tabulated at each site to enable the description of the recruitment process, strategies and outcomes. Cross-site analysis then compared these findings.

## FINDINGS

### Site characteristics

Site A served four diverse London boroughs. Site B was a Northern city with a predominately urban population. Site C served an urban and rural population in the North of England. Sites A and C had experience of recruiting to and delivery of psychosocial intervention dementia research. This was the first large-scale psychosocial intervention dementia study site B had participated in.

### Recruitment targets

Recruitment targets for each site were based on the findings of a pilot study at each of these three sites which examined the feasibility of study procedures and recruitment. This, as well as investigator experiences of successful recruitment to psychosocial dementia research and numbers of occupational therapists trained and available to deliver the intervention, indicated the numbers each site could be expected to recruit. The number, type and experience of staff dedicated to recruitment varied at each site.

Sites A, B and C had targets of 90, 80 and 60, respectively. Initially, recruitment was scheduled for 18 months but was extended when recruitment proved slower than anticipated. As shown in table 1, site C recruited over the agreed target within the planned time frame. In comparison, sites A and B took 18 months longer to recruit 92% and 91%, respectively, of their target numbers.

### Identified recruitment strategies

The differences and similarities in recruitment strategies at the three sites are summarised in table 2. Similar strategies were employed at all three sites with National Health Service (NHS) memory services being the main source of participants at each. Memory services provide specialist diagnostic services and postdiagnostic support. At sites A and B, but not at site C, researchers maintained a regular presence in memory service clinics so that they were readily available to talk to potential recruits. At site B, a research nurse also prescreened clinical records to identify potentially eligible people to memory service clinicians in advance of routine appointments. Also, at this site, only study information was displayed at general practitioner (GP) practices at which this memory service offered postdiagnostic follow-up appointments. At site C, multidisciplinary clinical team meetings were used to identify potential recruits, this was not noted at the other two sites. At sites A and C, recruitment was extended into the non-statutory sector (charities and organisations supporting people affected by dementia). A further

**Table 1** Recruitment targets, number of potential dyads, number of dyads consented, percentage of target achieved, and time taken, by site

|  | Site A | Site B | Site C |
|---|---|---|---|
| Target (dyads) | 90 | 80 | 60 |
| Potential dyads identified during study recruitment | 332 | 233 | 144 |
| Total number consented (% of potential dyads identified that 'converted' into participants) | 83 (25%) | 73 (31%) | 73 (51%) |
| % of target achieved | 92% | 91% | 122% (*13 dyads above target*) |
| Length of time taken to recruit (months) | 29 | 29 | 11 |

NB. percentages rounded to the nearest whole number

**Table 2** Recruitment strategies used to identify potential participants, by site

| | Site A | Site B | Site C |
|---|---|---|---|
| **Recruitment strategy** | | | |
| *Within NHS site memory services* | | | |
| Direct referral by memory services clinicians | ✔ | ✔ | ✔ |
| Regular presence in memory services clinics by researchers | ✔ | ✔ | – |
| Attendance at psychosocial intervention groups by researchers | ✔ | ✔ | ✔ |
| 'Pre' screening of clinical records by a research nurse | – | ✔ | – |
| Leaflets and posters displayed | ✔ | ✔ | ✔ |
| Ad-hoc mail outs targeting potentially eligible participants choosing to attend follow-up appointments offered at local GP practices, instead of memory services at the hospital | – | ✔ | – |
| *Within other services provided by the NHS site* | | | |
| Potential participants identified by within multidisciplinary clinical meetings | – | – | ✔ |
| Occupational therapists delivering the intervention identifying potential participants | ✔ | ✔ | ✔ |
| Attendance at clinical team business meetings by researchers | ✔ | – | ✔ |
| Leaflets and posters displayed (other NHS Trust locations) | – | ✔ | ✔ |
| Research team made contact with people who had participated in other studies previously and had agreed to be contacted about future studies | ✔ | – | ✔ |
| *Involvement of other NHS providers* | | | |
| Information displayed in GP practices associated with memory services | – | ✔ | – |
| Patient Identification Centre in another NHS Trust | – | ✔ | – |
| *Non-NHS* | | | |
| Attendance at community groups by research staff | ✔ | – | ✔ |
| Study promoted by researchers at local events | ✔ | ✔ | ✔ |
| One mail out via non-statutory sector organisation / sending non-statutory sector organisation staff study information | ✔ | – | ✔ |
| 'Join Dementia Research' [1] (JDR) (https://www.joindementiaresearch.nihr.ac.uk): an online resource that enables people to register interest in participating in dementia research and thereby be 'matched' to relevant studies. Researchers then contact them directly. People who expressed interest living within the sites' locality, were sent information when JDR became active at each site. | ✔ | ✔ | ✔ |

GP, general practitioner; NHS, National Health Service.

strategy, at sites A and C, was to contact eligible people who had taken part in previous dementia research studies. Site C identified 15 additional potential recruits this way and site A identified 5. This route was not available in site B as no dementia research whereby people with dementia were asked for consent to be contacted about other research studies had taken place .

### Reasons for exclusion

The main reasons documented for exclusion are presented in table 3. The two sites which took the longest to recruit their target numbers (A and B) also had larger numbers of people excluded due to being ineligible or unwilling to participate.

As table 4 shows, reasons given by those unwilling or unable to participate (when provided) were recorded at all sites. It was not possible from the available records to determine if it was the person with dementia, the family carer or both members of the dyad who declined to participate. The numbers of potential participants excluded due to being unable or unwilling to participate at sites A and B outnumbered those excluded for all other reasons including individuals that researchers had been unable to contact. Site C recorded the lowest number of people being unable or unwilling to participate. The 'other reasons' for the declining category included adjusting to the dementia diagnosis, participation being perceived to

**Table 3** Main reasons recorded by research staff for exclusion by site

| Reason for exclusion | Site A | Site B | Site C |
|---|---|---|---|
| Study inclusion criteria not met or exclusion criteria identified* | 53 (21%) | 18 (11%) | 8 (11%) |
| No contact made | 64 (26%) | 28 (18%) | 12 (17%) |
| 'Unable or unwilling to participate' recorded as reason | 132 (53%) | 114 (71%) | 51 (72%) |
| Total excluded | 249 | 160 | 71 |

*Inclusion criteria not met/exclusion criteria identified included person with dementia not living in the community, not having capacity to consent, not score 0.5–2 on clinical dementia rating scale[18] or no family carer available to participate, a dyad participated in an earlier phase of the study or was participating in another intervention study or was unable to communicate fluently in English.

be a burden and personal circumstances (such as travel plans, moving house or bereavement).

## DISCUSSION

This secondary analysis of routinely collected recruitment data for one study involving people with dementia was highly informative. We found that successful recruitment of people with dementia, as one example of a hard to reach group, requires multiple strategies and necessitates close working between researchers and clinical services. All sites found recruitment to this psychosocial intervention study to be challenging, but one site did achieve the target numbers of participants within the allocated time.

**Table 4** Recorded explanations for being unable or unwilling to participate, by site

| Main explanation (if given) for being unable or unwilling to participate | Site A | Site B | Site C |
|---|---|---|---|
| Declined participation, no reason recorded | 53 (40%) | 50 (44%) | 12 (24%) |
| Physical ill health of either person | 6 (5%) | 15 (13%) | 4 (8%) |
| 'Managing fine' reported | 11 (8%) | 6 (5%) | 11 (22%) |
| Time constraints reported | 25 (19%) | 24 (21%) | 11 (22%) |
| Other reasons recorded | 37 (28%) | 19 (17%) | 13 (25%) |
| Total potential participants recorded as unable or unwilling to participate | 132 | 114 | 51 |

NB percentages rounded to the nearest whole number.

Our findings showed the original pool of people available for recruitment quickly funnelled to a few at each site for a variety of reasons. Initial optimism regarding the potential pool of participants was fuelled by optimistic clinician estimates and our desire as researchers to be persuaded by these figures. It was also underscored by the need to work within the limitations set by the funder, as a better recruitment rate would be less costly and contribute towards a successful study. Alternatively, less optimistic recruitment estimates would raise doubts about study viability. This poses questions about how researchers can realistically estimate the recruitment efforts required for any study. We would like to encourage debate about this issue.

### Novel contribution

We interrogated the challenges of recruiting to one dementia study and argue for routine sharing of such experiences between researchers. We identified several key issues that appeared to affect recruitment in this study, which are likely to have implications for research conducted with other hard to reach groups. Possible reasons for recruitment challenges are organisational and individual.

### Organisational factors

Research site experience of recruitment to and running similar studies appears to be a critical issue. The exemplar in this paper involved recruiting a hard to reach population to a complex psychosocial intervention study, which potentially required significant time investment by participants. Although sites A and C had established working relationships between clinicians and site-based researchers, the most successful recruitment site (C) was also able to identify potentially eligible participants within multidisciplinary clinical meetings. This demonstrated active rather than passive clinical engagement in the study and consequently, the identification of those who were most appropriate to approach. Due to previous experience of running such studies, both sites A and C approached people who had previously consented to be contacted for potential participation in other studies as one of their strategies. Site C was able to approach greater numbers this way. For site B, this was not possible. Staff at sites A and C were both experienced in delivering psychosocial intervention dementia research but recruited at different rates which was not expected. This analysis confirms that no single factor is responsible for recruitment, rather effective recruitment depends on the interplay between a combination of factors. Different populations, demographics or research fatigue may have influenced the different recruitment outcomes. The number of other research studies running at sites may also have affected the engagement of NHS research and clinical services. An additional factor affecting recruitment for psychosocial intervention studies such as this is the requirement for staff to deliver the intervention. The recruitment of participants has to be matched with

the availability and sustainability of this workforce. In this study, sometimes recruitment at sites was temporarily halted until an occupational therapist was available to deliver the intervention. It is well known that clinical staff can act as gatekeepers, they may be unclear about the benefits of research projects, worry about overburdening patients or fear patients may feel pressurised to participate.[6] However, it seems that site C managed to overcome these issues and make research a positive aspect of clinical care.

### Individual factors

Alongside these organisational challenges, individual factors affected the responses of potential participants. It is well documented that people with dementia can be hard to reach.[4 7 9] Various reasons for this have been identified, including family carers wishing to protect people with dementia from potentially stressful situations or burden.[10] Although we could not determine whether this was the case from the available data, it seems likely this could be a contributory factor to recruitment challenges, for example; some of the records examined noted family carers reporting the person with dementia did not accept their diagnosis, or became upset when dementia was mentioned. Linguistic difficulties or people with dementia lacking capacity to consent have also been noted as reasons that can lead to recruitment difficulties for this hard to reach group[4] and in this exemplar, people with dementia were excluded for those reasons. Researcher notes indicated some potential participants reported they had 'too much on', suggesting participation was perceived by some as burdensome without offering enough potential benefit to compensate for this. Other researchers have found that studies can be perceived as time-consuming particularly for adult children or that people with dementia may be concerned about burdening relatives with the role of study partner.[11 12] Reasons recorded for declining may also have been polite refusals obscuring other reasons for declining which remain unknown. The message here is that researchers need to understand and be able to respond appropriately to the needs and preferences of the specific hard to reach group. Generic research training is not sufficient.

### Possible recruitment solutions

We suggest the following as potential strategies to improve recruitment efforts for future research studies involving hard to reach populations.

First, making the potential benefits of research transparent to potential participants is important, as is the involvement of clinical services and family carers. Law et al[13] found that people with dementia wanted to be asked directly and involvement in research can lead to feeling valued and sense of being able to contribute. Asking the person with the condition directly about their potential involvement, if they have the capacity to provide this, is essential. As our findings demonstrate, there are also advantages in ensuring that relevant services are on board

and perceive engagement in the research to be relevant to them and the people that they work with. But, as Iliffe et al[14] noted, the need to support research infrastructure for psychosocial dementia research remains.

Second, a national research registry whereby people with dementia and caregivers are asked for consent to be approached for research participation can help identify potential recruits.[7 15] Further, some NHS trusts in England are developing systems whereby patients can be asked for their permission to be contacted about research at any point in their care pathway. If staff are persuaded by the potential benefits of research, then this strategy may aid recruitment.

Third, transparent reporting of recruitment strategies and how many people were initially identified as being potentially eligible including the contexts within which recruitment took place will support knowledge sharing. Analysis of recruitment methods should ideally be built into study designs to allow detailed reflection as an intrinsic part of large studies involving hardtoreach groups. There is a need for research to examine the impact of the type of dementia diagnosis, age, comorbidities, socioeconomic status, ethnicity, education or type of caring relationship, as well as different recruitment methods, on participation or non-participation, in studies to further illuminate influences on recruitment.

The analysis used as an exemplar in this article was completed once the study had been designed and commenced and had limitations. For example, resources meant we were able to examine recruitment experiences at these three sites only, rather than all 15. Also, we cannot comment on the effectiveness of any single recruitment strategy used at each site or the relationship of key characteristics of participants on recruitment outcomes. Despite this, what we can say is that it seems an interplay of organisational and individual factors influenced recruitment outcomes and this needs to be considered in future studies. We contend that completing similar analyses as studies progress, if building this into the initial plan is not feasible, is still worthwhile. Such work can enable learning to be shared, across study sites and with other research teams.

Fourth, comprehensive researcher understanding of the perspectives and needs, including any special requirements of the specific hard to reach population being studied, is necessary. For example, identifying ways to engage people with cognitive impairments, perhaps alongside comorbidities, sensory and physical impairments that limit the social participation of people with dementia[16] may facilitate recruitment. Communication style is important and may need adapting.[17] This may well require additional researcher training and on-going support.

### CONCLUSIONS

Successful recruitment of people from hard to reach groups, such as people with dementia, requires multiple

strategies and necessitates close working between researchers and relevant services. It requires a detailed understanding of the needs and perspectives of the specific population and application of knowledge regarding how individuals can be supported to participate in research. Reporting the evaluation of recruitment strategies and experiences should be an expected output from large studies. This would enhance understanding about how to enable hard to reach populations to participate in studies.

**Author affiliations**
¹School of Health and Related Research, Faculty of Medicine Dentistry and Health, University of Sheffield, Sheffield, UK
²Centre for Applied Dementia Studies, University of Bradford, Bradford, UK
³Research and Development Department, North East London NHS Foundation Trust Goodmayes Hospital, Ilford, UK
⁴Engagement and Experience Team, Sheffield Health and Social Care NHS Foundation Trust, Sheffield, UK
⁵Research and Development Department, Humber Teaching NHS Foundation Trust, Willerby, UK
⁶Division of Psychiatry, University College London, London, UK

**Contributors** BF and GM: developed the study concept and design for this paper. BF: led data analysis and interpretation, assisted by GM, LDB, JM, JB, JW and DK; drafted the manuscript. All authors contributed to its revision and approved the final manuscript.

**Funding** This manuscript presents independent research funded by the National Institute for Health Research's Programme Grants for Applied Research Programme (RP-PG 0610-10108). The views expressed are those of the authors and not necessarily those of the NHS, the NIHR or the Department of Health.

**Competing interests** None declared.

**Patient consent for publication** Not required.

**Provenance and peer review** Not commissioned; externally peer reviewed.

**ORCID iD**
Becky Field http://orcid.org/0000-0003-3502-2691

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
