## [Reviewer comments · BMJ Open]

ARTICLE DETAILS

TITLE (PROVISIONAL)	Recruiting hard to reach populations to studies: breaking the silence. An example from a study that recruited people with dementia.
AUTHORS	Field, Becky; Mountain, Gail; Burgess, Jane; Di Bona, Laura; Kelleher, Daniel; Mundy, Jacqueline; Wenborn, Jennifer

VERSION 1 – REVIEW

REVIEWER	Jonas Ranstam Lund University, Sweden
REVIEW RETURNED	05-Jun-2019

GENERAL COMMENTS	The reviewer completed the checklist but made no further comments.
--

REVIEWER	Jeremy M. Silverman Icahn School of Medicine at Mount Sinai Department of Psychiatry New York, NY USA
REVIEW RETURNED	10-Jun-2019

GENERAL COMMENTS	The paper asks the question, "What subject recruitment strategies are efficacious for dementia research?" and, as an exemplar, reports on the recruitment experience of a recent multi-site investigation. As noted in the paper, this question is rarely discussed despite its critical importance for clinical dementia research. The paper is clearly written and identifies several issues that could be helpful for research planning, but there are a number of areas not discussed that if provided might help further: 1. While this a paper about recruitment and does not address the actual study question involved, it would nevertheless be helpful to know in a sentence or two what the research study was about and what it entailed. Obviously, the purpose of the study along with the specific involvement and activities being asked of the subjects and their caregivers will affect enrollment.2. Site A's initial target of 90 of 332 potential dyads required a 27% success rate, Site B's 80 of 233 required a 34% success rate and Site C's target of 60 of 144 potential dyads required a 42% success rate. How were these rates determined for the different sites? It would appear that from the start, Site C was anticipated to be the most successful as in fact it was. Why?3. The sites are kept anonymous in the paper, but the characteristics of the populations they serve may well be associated with recruitment success. Do the sites differ, for example, with respect to their respective community's socio-
--

	economic status? their location (urban versus rural status)? or the proportion of minority, ethnic, or linguistic groups? 4. As noted, owing to the frequent incapacity associated with dementia for informed consent, recruitment usually requires the involvement of a cooperative caregiver. Were their differences between sites by sex or by the relationship status of the caregiver (husband, wife, son, daughter, etc.). 5. If there is information available about the potential dyads identified who did not ultimately enroll in the study, it would be of interest to compare demographics between participants and non-participants overall and whether the three sites varied on these characteristics.
--	--

REVIEWER	Shoshana Bardach University of Kentucky, USA
REVIEW RETURNED	18-Jun-2019

GENERAL COMMENTS	This article explores an important topic, recruitment research for dementia studies. This is a challenging area to study and the authors did a nice job. My suggestions for improvement are as follows:  1) Page 2 line 57- "necessitates close working" seems to be missing a word. 2) The decision to participate in research is influenced by the research itself. More information about the intervention of focus is needed, e.g. what was the time commitment for participation? What were the procedures like? Where did it occur? These details should be included on page 4. 3) It's unclear to me why you chose to only focus on 3 of the 15 participating sites. It seems like you could likely make far more conclusions by looking at all 15. Can you explain why you chose to focus just on 3? Also, more description of the 3 sites is needed. E.g. what type of sites were they? Did they typically do dementia research? Did they have other studies going on? Did they differ in reputation? Did their contact strategies differ? The written description would also flow better if you describe the target numbers and time frame before reporting the results. Also, more information regarding whether there were delays initially, throughout, etc., would also be helpful. 4) Capturing data during the recruitment process often seems to vary site-to-site, can you describe the procedures used at the studied sites so that this information was available for analysis? Was there a standard way this was tracked across sites? Who were the personnel involved in recruitment and did the number vary across sites? 5) Can you define "memory services"? It's not entirely clear what this refers to. Definitions would also be helpful for NHS and statutory sector. 6) Tables 3 and 4 would benefit from the inclusion of percentages. 7) The discussion seems to speak beyond the power of the study. For instance, the authors make conclusions about the effectiveness of different strategies on page 9, but have not previously reported analysis of individual strategies. Accordingly, it doesn't seem possible to conclude which of many approaches accounted for differences between sites. If there was a way to discern this it needs to be discussed. If not, these conclusions need to be removed or at least reworded. 8) Page 1- line 20, "too much on", should this be "too much going on"?
---

	9) While recruitment research is an emerging area, the discussion is rather under-referenced. I would encourage the authors to consider incorporating additional references into this discussion. In addition, while other hard to reach groups are also important, I would caution against making conclusions about other populations based on this one effort.
--	--

VERSION 1 – AUTHOR RESPONSE

Reviewer 2 comments	Author's response
1. While this a paper about recruitment and does not address the actual study question involved, it would nevertheless be helpful to know in a sentence or two what the research study was about and what it entailed. Obviously, the purpose of the study along with the specific involvement and activities being asked of the subjects and their caregivers will affect enrolment.	We have edited the introduction to clarify this and the reference given provides further details (p.5 lines 21-31)
2. Site A's initial target of 90 of 332 potential dyads required a 27% success rate, Site B's 80 of 233 required a 34% success rate and Site C's target of 60 of 144 potential dyads required a 42% success rate. How were these rates determined for the different sites? It would appear that from the start, Site C was anticipated to be the most successful as in fact it was. Why?	Site specific targets of 90, 80 and 60 were identified at the outset, rather than being calculated as a percentage of number of potential dyads identified. The different targets were determined after a pilot study run at these three sites, the basis of investigator experience of running psychosocial dementia research studies and number of occupational therapists available. We have edited this section and Table 1, to improve clarity as the reviewer comments indicate that our initial explanation was unclear (p6 lines 6-9 and table 1 page 7 lines 9-11) It was not anticipated that Site C would be more successful than the other two sites. But as reported, Site C did recruit most effectively. We consider potential reasons for this in the discussion under 'organisational factors. We have edited this section to take on board reviewer 2 comments. Please see page 11 lines 27-8 , 32-41 p12 lines1-6

3. The sites are kept anonymous in the paper, but the characteristics of the populations they serve may well be associated with recruitment success. Do the sites differ, for example, with respect to their respective community's socio-economic status? their location (urban versus rural status)? or the proportion of minority, ethnic, or linguistic groups?	We agree characteristics of the populations served by each site may well be associated with recruitment success. We have edited the paper to include a brief description of each site (p6 lines 23-28). The aim of this short 'communication' paper was to share challenges experienced and lessons learnt. We did not aim to examine relationships / associations of specific characteristics on recruitment as part of this secondary analysis. Of course such primary research would be valuable and we believe merits a primary analysis, of data collected for that purpose. We have edited the discussion to reflect this issue and the importance of examining impact of key characteristics on recruitment (p13 lines 18-21).
4. As noted, owing to the frequent incapacity associated with dementia for informed consent, recruitment usually requires the involvement of a cooperative caregiver. Were their differences between sites by sex or by the relationship status of the caregiver (husband, wife, son, daughter, etc.).	As the aim of this paper was to share challenges experienced and lessons learnt, we did not examine this as part of this secondary analysis. All sites did collect data about caregiver relationship and gender, this data could be available for further analysis in future. We have edited the discussion to reflect the importance of examining the impact of such characteristics on recruitment (p13 lines 18-21).
5. If there is information available about the potential dyads identified who did not ultimately enroll in the study, it would be of interest to compare demographics between participants and non-participants overall and whether the three sites varied on these characteristics.	Very limited information about the dyads who did not participate is available. A small telephone interview study completed with carers about reasons for non-participation was conducted within the main study, but this is yet to be published. We have edited the discussion to reflect the need to consider reasons for non-participation (p.13 line20-21)
Reviewer 3 comments	
1) Page 2 line 57- "necessitates close working" seems to be missing a word.	We have accommodated this comment by removing the word 'necessitates' (p3 line 31)
2) The decision to participate in research is influenced by the research itself. More information about the intervention of focus is needed, e.g. what was the time commitment for participation? What were the procedures like? Where did it occur? These details should be included on page 4.	We have accommodated this comment and edited to include further details of the study people were recruited to and a reference for further details (p5 lines_21-31)
3) It's unclear to me why you chose to only focus on 3 of the 15 participating sites. It seems like you could likely make far more conclusions by looking at	We edited p.6 lines_3-9 to clarify that these three sites were samples as they had been the sites for the pilot study which identified substantial numbers of potential

all 15. Can you explain why you chose to focus just on 3? -Also, more description of the 3 sites is needed. E.g. what type of sites were they? Did they typically do dementia research? Did they have other studies going on? Did they differ in reputation? Did their contact strategies differ? The written description would also flow better if you describe the target numbers and time frame before reporting the results. Also, more information regarding whether there were delays initially, throughout, etc., would also be helpful.	participants at each of these sites, and that these sites had the resources to support the secondary analysis. We agree more conclusions could be reached if all 15 sites' recruitment data had been examined. But, as this was a secondary analysis and not one initially planned for or costed for at the outset for by the main study we were unable to examine data from all 15 sites. Rather our aim was to share challenges and lessons learnt, focusing on these three sites. One lesson is certainly to build in analysis or reflection of recruitment into study design if possible, as noted in the discussion (p13 lines14-18).We have also added that we were only able to examine 3 of 15 sites recruitment data (p13, line 23-4). We have edited the Findings section to provide more information about the sites, clarify whether they typically did dementia research (p.6 lines 24-28) initial time frame for recruitment (p7 lines 1-3) and give a written description of target numbers and time frame before reporting the results, as suggested (p7 lines 1-5). We were uncertain what 'reputation' and 'contact strategies' meant in this context. The different recruitment strategies used in each site, including strategies for making contact with potential participants are presented in Table 2 page 9. The increased detail we have provided about the sites, summarises the previous relevant research carried out at each site. We have edited the discussion to provide more information as suggested, about delays affecting recruitment and the differences between sites in terms of experience of psychosocial intervention dementia (p.11 lines 32-41, p12 1-6)
4) Capturing data during the recruitment process often seems to vary site-to-site, can you describe the procedures used at the studied sites so that this information was available for analysis? Was there a standard way this was tracked across sites? Who were the personnel involved in recruitment and did the number vary across sites?	The data collected during the recruitment process did vary, to some extent, at each site. Each site collected some 'core' information to enable screening for study inclusion criteria. This was collected in a 'trial management log' by each site. A template was provided but could be adapted as needed to meet local circumstances e.g. to facilitate communication between staff, with potential recruits/ booking appointments. All sites recorded reasons why people declined, but the way this was done locally varied across these three sites. These researcher notes recording reasons given

	for declining or ineligibility were then grouped into categories of reasons for declining or ineligibility. Site C used a different database to manage several dementia research studies and as the key information required for this study was contained on their own database, they used that. We have edited the Methods section to further clarify methods of data collection and analysis (page 6 lines 11-20). We acknowledge the type of number of staff at each site varied and this has now been included (p.6 36-7). Some further information about differing staff involvement at sites is also presented under the sub-heading 'identified recruitment strategies' (p.7 lines 16-21).
5) Can you define "memory services"? It's not entirely clear what this refers to. Definitions would also be helpful for NHS and statutory sector.	We have accommodated these comments (p7 lines 15-16 and 24)
6) Tables 3 and 4 would benefit from the inclusion of percentages.	We have accommodated these comments (p10) .Percentages were not included at submission as we felt percentages for such small numbers are not be helpful. We are happy for them to included or removed according to house style or editorial preference. .
7) The discussion seems to speak beyond the power of the study. For instance, the authors make conclusions about the effectiveness of different strategies on page 9, but have not previously reported analysis of individual strategies. Accordingly, it doesn't seem possible to conclude which of many approaches accounted for differences between sites. If there was a way to discern this it needs to be discussed. If not, these conclusions need to be removed or at least reworded.	We appreciate this point and have moderated and edited our discussion accordingly (p12-13 overall and particularly p13 lines23-27).
8) Page 1- line 20, "too much on", should this be "too much going on"?	We have left this as it is as it a colloquial phrase, (a comment from several potential participants recorded by researchers). Perhaps this is a difference between American and English?
9) While recruitment research is an emerging area, the discussion is rather under-referenced. I would encourage the authors to consider incorporating additional references into this discussion. In addition, while other hard to reach groups are also important, I would caution against making conclusions about other populations based on this one effort	We have included five more references to support our discussion and have addressed these comments. As this is not a traditional research article we thought it important to try and appeal to readers who may work with other hard to reach groups, not just dementia. Of course we do not wish to make conclusions about other populations based on this one effort. To clarify this issue we have edited the discussion (p13, lines 7-13, 18-21, 33-36) to focus on the specific population we researched.